# Q-Value Weighted Regression: Reinforcement Learning with Limited Data

## Abstract

Sample efficiency and performance in the offline setting have emerged as among the main challenges of deep reinforcement learning. We introduce Q-Value Weighted Regression (QWR), a simple RL algorithm that excels in these aspects. QWR is an extension of Advantage Weighted Regression (AWR), an off-policy actor-critic algorithm that performs very well on continuous control tasks, also in the offline setting, but struggles on tasks with discrete actions and in sample efficiency. We perform a theoretical analysis of AWR that explains its shortcomings and use the insights to motivate QWR theoretically. We show experimentally that QWR matches state-of-the-art algorithms both on tasks with continuous and discrete actions. We study the main hyperparameters of QWR and find that it is stable in a wide range of their choices and on different tasks. In particular, QWR yields results on par with SAC on the MuJoCo suite and – with the same set of hyperparameters – yields results on par with a highly tuned Rainbow implementation on a set of Atari games. We also verify that QWR performs well in the offline RL setting, making it a compelling choice for reinforcement learning in domains with limited data.

## 1 Introduction

Deep reinforcement learning has been applied to a large number of challenging tasks, from games (Silver et al., 2017; OpenAI, 2018; Vinyals et al., 2017) to robotic control (Sadeghi & Levine, 2016; OpenAI et al., 2018; Rusu et al., 2016). Since RL makes minimal assumptions on the underlying task, it holds the promise of automating a wide range of applications. However, its widespread adoption has been hampered by a number of challenges. Reinforcement learning algorithms can be substantially more complex to implement and tune than standard supervised learning methods and can have a fair number of hyper-parameters and be brittle with respect to their choices, and may require a large number of interactions with the environment.

These issues are well-known and there has been significant progress in addressing them. The policy gradient algorithm REINFORCE (Williams (1992)) is simple to understand and implement, but is both brittle and requires on-policy data. Proximal Policy Optimization (PPO, Schulman et al. (2017)) is a more stable on-policy algorithm that has seen a number of successful applications despite requiring a large number of interactions with the environment. Soft Actor-Critic (SAC, Haarnoja et al. (2018)) is a much more sample-efficient off-policy algorithm, but it is defined only for continuous action spaces and does not work well in the offline setting, known as batch reinforcement learning, where all samples are provided from earlier interactions with the environment, and the agent cannot collect more samples. Advantage Weighted Regression (AWR, Peng et al. (2019)) is a recent off-policy actor-critic algorithm that works well in the offline setting and is built using only simple and convergent maximum likelihood loss functions, making it easier to tune and debug. It is competitive with SAC given enough time to train, but is less sample-efficient and has not been demonstrated to succeed in settings with discrete actions.

We replace the value function critic of AWR with a Q-value function. Next, we add action sampling to the actor training loop. Finally, we introduce a custom backup to the Q-value training. The resulting algorithm, which we call Q-Value Weighted Regression (QWR) inherits the advantages of AWR but is more sample-efficient and works well with discrete actions and in visual domains, e.g., on Atari games.

To better understand QWR we perform a number of ablations, checking different number of samples in actor training, different advantage estimators, and aggregation functions. These choices affect the performance of QWR only to a limited extent and it remains stable with each of the choices across the tasks we experiment with.

We run experiments with QWR on the MuJoCo environments and on a subset of the Atari Learning Environment. Since sample efficiency is our main concern, we focus on the difficult case when the number of interactions with the environment is limited – in most our experiments we limit it to 100K interactions. The experiments demonstrate that QWR is indeed more sample-efficient than AWR. On MuJoCo, it performs on par with Soft Actor-Critic (SAC), the current state-of-the-art algorithm for continuous domains. On Atari, QWR performs on par with OTRainbow, a variant of Rainbow highly tuned for sample efficiency. Notably, we use the same set of hyperparameters (except for the network architecture) for both our final MuJoCo and Atari experiments.

## 2  Q-VALUE WEIGHTED REGRESSION

### 2.1  ADVANTAGE WEIGHTED REGRESSION

Peng et al. (2019) recently proposed Advantage Weighted Regression (AWR), an off-policy, actor-critic algorithm notable for its simplicity and stability, achieving competitive results across a range of continuous control tasks. It can be expressed as interleaving data collection and two regression tasks performed on the replay buffer, as shown in Algorithm 1.

---

**Algorithm 1** Advantage Weighted Regression.

1:  $\theta \leftarrow$ random actor parameters
2:  $\phi \leftarrow$ random critic parameters
3:  $\mathcal{D} \leftarrow \emptyset$
4:  **for** $k$ **in** $0..n\_iterations - 1$ **do**
5:      add trajectories $\{\tau_i\}$ sampled by $\pi_\theta$ to $\mathcal{D}$
6:      **for** $i$ **in** $0..n\_critic\_steps - 1$ **do**
7:          sample $(\mathbf{s}, \mathbf{a}) \sim \mathcal{D}$
8:          $\phi \leftarrow \phi - \alpha_V \nabla_\phi \left[ ||\mathcal{R}_\mathcal{D}^{\mathbf{s},\mathbf{a}} - V_\phi(\mathbf{s})||^2 \right]$
9:      **end for**
10:     **for** $i$ **in** $0..n\_actor\_steps - 1$ **do**
11:         sample $(\mathbf{s}, \mathbf{a}) \sim \mathcal{D}$
12:         $\theta \leftarrow \theta + \alpha_\pi \nabla_\theta \left[ \log \pi_\theta(\mathbf{a}|\mathbf{s}) \exp(\frac{1}{\beta}(\mathcal{R}_\mathcal{D}^{\mathbf{s},\mathbf{a}} - V_\phi(\mathbf{s}))) \right]$
13:     **end for**
14: **end for**

---

AWR optimizes *expected improvement* of an actor policy $\pi(\mathbf{a}|\mathbf{s})$ over a sampling policy $\mu(\mathbf{a}|\mathbf{s})$ by regression towards the well-performing actions in the collected experience. Improvement is achieved by weighting the actor loss by exponentiated advantage $A_\mu(\mathbf{s}, \mathbf{a})$ of an action, skewing the regression towards the better-performing actions. The advantage is calculated based on the expected return $\mathcal{R}_\mu^{\mathbf{s},\mathbf{a}}$ achieved by performing action $\mathbf{a}$ in state $\mathbf{s}$ and then following the sampling policy $\mu$. To calculate the advantage, one first estimates the value, $V_\mu(s)$, using a learned critic and then computes $A_\mu(\mathbf{s}, \mathbf{a}) = \mathcal{R}_\mu^{\mathbf{s},\mathbf{a}} - V_\mu(\mathbf{s})$. This results in the following formula for the actor:

$$\arg\max_\pi \mathbb{E}_{\mathbf{s} \sim d_\mu(\mathbf{s})} \mathbb{E}_{\mathbf{a} \sim \mu(\cdot|\mathbf{s})} \left[ \log \pi(\mathbf{a}|\mathbf{s}) \exp\left( \frac{1}{\beta}(\mathcal{R}_\mu^{\mathbf{s},\mathbf{a}} - V_\mu(\mathbf{s})) \right) \right], \tag{1}$$

where $d_\mu(\mathbf{s}) = \sum_{t=1}^{\infty} \gamma^{t-1} p(\mathbf{s}_t = \mathbf{s}|\mu)$ denotes the unnormalized, discounted state visitation distribution of the policy $\mu$, and $\beta$ is a temperature hyperparameter.

The critic is trained to estimate the future returns of the sampling policy $\mu$:

$$\arg\min_V \mathbb{E}_{\mathbf{s} \sim d_\mu(\mathbf{s})} \mathbb{E}_{\mathbf{a} \sim \mu(\cdot|\mathbf{s})} \left[ ||\mathcal{R}_\mu^{\mathbf{s},\mathbf{a}} - V(\mathbf{s})||^2 \right]. \tag{2}$$

To achieve off-policy learning, the actor and the critic are trained on data collected from a mixture of policies from different training iterations, stored in the replay buffer $\mathcal{D}$.

## 2.2 ANALYSIS OF AWR WITH LIMITED DATA

While AWR achieves very good results after longer training, it is not very sample efficient, as noted in the future work section of (Peng et al., 2019). To understand this problem, we analyze a single loop of actor training in AWR under a special assumption.

The assumption we introduce, called *state-determines-action*, concerns the content of the replay buffer $\mathcal{D}$ of an off-policy RL algorithm. The replay buffer contains all state-action pairs that the algorithm has visited so far during its interactions with the environment. We say that a replay buffer $\mathcal{D}$ satisfies the *state-determines-action* assumption when for each state $s$ in the buffer, there is a unique action that was taken from it, formally:

$$\text{for all } (s, a), (s', a') \in \mathcal{D} : s = s' \implies a = a'.$$

This assumption may seem very limiting and indeed – it is not true in many artificial experiments with RL algorithms, such as Atari games. Even a random policy starting from the same state could violate the assumption the second time it collects a trajectory. But note that this assumption is almost always satisfied in real-world experiments with high-dimensional state spaces as any amount of noise added to a high-dimensional space will make repeating the exact same state highly improbable. For example, consider a robot observing 32x32 pixel images. To repeat a state, each of the 1024 pixels would have to have exactly the same value, which is highly improbable even with a small amount of noise. This assumption also holds in cases with limited data. When only a few trajectories are collected in a large state space, it is unlikely a state will be repeated in the replay buffer, which makes the assumption relevant to the study of sample efficiency.

How does AWR perform under the state-determines-action assumption? In Theorem 1, formulated and proven in Appendix A.2, we show that the AWR update rule under this assumption will converge to a policy that assigns probability 1 to the actions already present in the replay buffer, thus cloning the previous behaviors. This is not the desired behavior, as an agent should consider various actions from each state, to ensure exploration.

The state-determines-action assumption is the main motivating point behind QWR, whose theoretical properties are proven in Theorem 2 in Appendix A.3. We now illustrate the importance of this assumption by creating a simple environment in which it holds with high probability. We verify experimentally that AWR fails on this simple environment, while QWR is capable of solving it.

The environment, which we call *BitFlip*, is parameterized by an integer $N$. The state of the environment consists of $N$ bits and a step counter. The action space consists of $N$ actions. When an action $i$ is chosen, the $i$-th bit is flipped and the step counter is incremented. A game of BitFlip starts in a random state with the step counter set to 0, and proceeds for 5 steps. The initial state is randomized in such a way to always leave at least 5 bits set to 0. At each step, the reward is $+1$ if a bit was flipped from 0 to 1 and the reward is $-1$ in the opposite case.

Since BitFlip starts in one random state out of $2^N$, at large enough $N$ it is highly unlikely that the starting state will ever be repeated in the replay buffer. As the initial policy is random and BitFlip maintains a step counter to prevent returning to a state, the same holds for subsequent states.

BitFlip is a simple game with a very simple strategy, but the initial replay buffer will satisfy the state-determines-action assumption with high probability. As we will see, this is enough to break AWR.

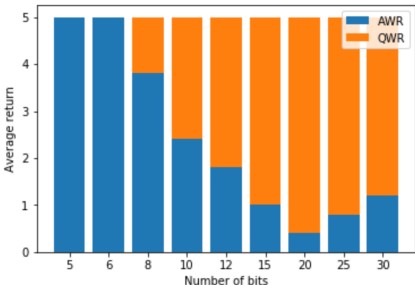

*Figure 1: AWR and QWR on the BitFlip environment. The maximum possible return is 5.*

We ran both AWR and QWR on BitFlip for different values of $N$, for 10 iterations per experiment. In each iteration we collected 1000 interactions with the environment and trained both the actor and the critic for 300 steps. All shared hyperparameters of AWR and QWR were set to the same values, and the backup operator in QWR was set to mean. We report the mean out of 10 episodes played by the trained agent. The results are shown in Figure 1.

As we can see, the performance of AWR starts deteriorating at a relatively small value of $N = 8$, which corresponds to a state space with $5 \cdot 2^8 = 1280$ states, while QWR maintains high performance even at $N = 30$, so around $5 \cdot 10^9$ states. Notice how the returns of AWR drop with $N$ – at higher values: $20 - 30$, the agent struggles to flip even a single zero bit. This problem with AWR and large state spaces motivates us to introduce QWR next.

## 2.3 Q-Value Weighted Regression

To remedy the issue indicated by Theorem 1, we introduce a mechanism to consider multiple different actions that can be taken from a single state. We calculate the advantage of the sampling policy $\mu$ based on a learned Q-function: $A_\mu(\mathbf{s}, \mathbf{a}) = Q_\mu(\mathbf{s}, \mathbf{a}) - \hat{V}_\mu(\mathbf{s})$, where $\hat{V}_\mu(\mathbf{s})$ is the expected return of the policy $\mu$, expressed using $Q_\mu$ by expectation over actions: $\hat{V}_\mu(\mathbf{s}) = \mathbb{E}_{a\sim\mu(\cdot|\mathbf{s})}Q_\mu(\mathbf{s}, \mathbf{a})$. We substitute our advantage estimator into the AWR actor formula (Equation 1) to obtain the QWR actor:

$$\arg\max_\pi \mathbb{E}_{s\sim d_\mu(\mathbf{s})}\mathbb{E}_{\mathbf{a}\sim\mu(\cdot|\mathbf{s})}\left[\log\pi(\mathbf{a}|\mathbf{s})\exp\left(\frac{1}{\beta}(Q_\mu(\mathbf{s}, \mathbf{a}) - \hat{V}_\mu(\mathbf{s}))\right)\right].\tag{3}$$

Similar to AWR, we implement the expectation over states in Equation 3 by sampling from the replay buffer. However, to estimate the expectation over actions, we average over multiple actions sampled from $\mu$ during training. Because the replay buffer contains data from multiple different sampling policies, we store the parameters of the sampling policy $\mu(\mathbf{a}|\mathbf{s})$ conditioned on the current state in the replay buffer and restore it in each training step to compute the loss. This allows us to consider multiple different possible actions for a single state when training the actor, not only the one performed in the collected experience.

The use of a Q-network as a critic provides us with an additional benefit. Instead of regressing it towards the returns of our sampling policy $\mu$, we can train it to estimate the returns of an improved policy $\mu^\star$, in a manner similar to Q-learning. This allows us to optimize expected improvement over $\mu^\star$, providing a better baseline - as long as $\mathbb{E}_{\mathbf{a}\sim\mu^\star(\cdot|\mathbf{s})}Q_\mu(\mathbf{s}, \mathbf{a}) \geq \mathbb{E}_{\mathbf{a}\sim\mu(\cdot|\mathbf{s})}Q_\mu(\mathbf{s}, \mathbf{a})$, the *policy improvement theorem* for stochastic policies (Sutton & Barto, 2018, Section 4.2) implies that the policy $\mu^\star$ achieves higher returns than the sampling policy $\mu$:

$$\mathbb{E}_{\mathbf{a}\sim\mu^\star(\cdot|\mathbf{s})}Q_\mu(\mathbf{s}, \mathbf{a}) \geq V_\mu(\mathbf{s}) \Rightarrow V_{\mu^\star}(\mathbf{s}) \geq V_\mu(\mathbf{s})\tag{4}$$

$\mu^\star$ need not be parametric - in fact, it is not materialized in any way over the course of the algorithm. The only requirement is that we can estimate the Q backup $\mathbb{E}_{\mathbf{a}\sim\mu^\star(\cdot|\mathbf{s})}Q(\mathbf{s}, \mathbf{a})$. This allows great flexibility in choosing the form of $\mu^\star$. Since we want our method to work also in continuous action spaces, we cannot compute the backup exactly. Instead, we estimate it based on several samples from the sampling policy $\mu$. Our backup has the form $\mathbb{E}_{\mathbf{a}_1,...,\mathbf{a}_k\sim\mu(\cdot|\mathbf{s})}F(\{Q(\mathbf{s}, \mathbf{a}_1), ..., Q(\mathbf{s}, \mathbf{a}_k)\})$. In this work, we extend the term *Q-learning* to mean training a Q-value using such a generalized backup. To make training of the Q-network more efficient, we use multi-step targets, described in detail in Appendix A.4. The critic optimization objective using single-step targets is:

$$\arg\min_Q \mathbb{E}_{\mathbf{s}\sim d_\mu(\mathbf{s})}\mathbb{E}_{\mathbf{a}\sim\mu(\mathbf{a}|\mathbf{s})}\mathbb{E}_{\mathbf{s}'\sim\mathcal{T}(\cdot|\mathbf{s},\mathbf{a})}\mathbb{E}_{\mathbf{a}'_1,...,\mathbf{a}'_k\sim\mu(\cdot|\mathbf{s}')}\left[||Q^\star - Q(\mathbf{s}, \mathbf{a})||^2\right],$$
$$\text{where } Q^\star = r(\mathbf{s}, \mathbf{a}) + \gamma F(\{Q_\mu(\mathbf{s}, \mathbf{a}_1), ..., Q_\mu(\mathbf{s}, \mathbf{a}_k)\})\tag{5}$$
$$\text{and } \mathcal{T}(\mathbf{s}'|\mathbf{s}, \mathbf{a}) \text{ is the environment's transition operator.}$$

In this work, we investigate three choices of $F$: average, yielding $\mu^\star = \mu$; max, where $\mu^\star$ approximates the greedy policy; and log-sum-exp, $F(X) = \tau\log\left[\frac{1}{|X|}\sum_{x\in X}\exp(x/\tau)\right]$, interpolating between average and max with the temperature parameter $\tau$. This leads to three versions of the QWR algorithm: QWR-AVG, QWR-MAX, and QWR-LSE. The last operator, log-sum-exp, is similar to the

backup operator used in maximum-entropy reinforcement learning (see e.g. Haarnoja et al. (2018)) and can be thought of as a soft-greedy backup, rewarding both high returns and uncertainty of the policy. It is our default choice and the final algorithm is shown in Algorithm 2.

---

**Algorithm 2** Q-Value Weighted Regression.

---

1: $\theta \leftarrow$ random actor parameters
2: $\phi \leftarrow$ random critic parameters
3: $\mathcal{D} \leftarrow \emptyset$
4: **for** $k$ **in** $0..n\_iterations - 1$ **do**
5:      add trajectories $\{\tau_i\}$ sampled by $\pi_\theta$ to $\mathcal{D}$
6:      $\phi_t \leftarrow \phi$
7:      **for** $i$ **in** $0..n\_critic\_steps - 1$ **do**
8:          **if** $i \mod update\_frequency = 0$ **then**
9:              $\phi_t \leftarrow \phi$
10:          **end if**
11:          sample $(\mathbf{s}, \mu, \mathbf{a}, \mathbf{r}, \mathbf{s}') \sim \mathcal{D}$
12:          sample $\mathbf{a}'_0, ..., \mathbf{a}'_{n-1} \sim \mu(\cdot|\mathbf{s}')$
13:          $Q^\star \leftarrow \mathbf{r} + \gamma F(\{Q_{\phi_t}(\mathbf{s}', \mathbf{a}'_0), ..., Q_{\phi_t}(\mathbf{s}', \mathbf{a}'_{n-1})\})$
14:          $\phi \leftarrow \phi - \alpha_V \nabla_\phi \left[ ||Q^\star - Q_\phi(\mathbf{s}, \mathbf{a})||^2 \right]$
15:      **end for**
16:      **for** $i$ **in** $0..n\_actor\_steps - 1$ **do**
17:          sample $(\mathbf{s}, \mu, ...) \sim \mathcal{D}$
18:          sample $\mathbf{a}_0, ..., \mathbf{a}_{n-1} \sim \mu(\cdot|\mathbf{s})$
19:          $\hat{V} \leftarrow \frac{1}{n} \sum_{j=0}^{n-1} Q_\phi(\mathbf{s}, \mathbf{a}_j)$
20:          $\theta \leftarrow \theta + \alpha_\pi \nabla_\theta \frac{1}{n} \sum_{j=0}^{n-1} \left[ \log \pi_\theta(\mathbf{a}_j|\mathbf{s}) \exp(\frac{1}{\beta}(Q_\phi(\mathbf{s}, \mathbf{a}_j) - \hat{V})) \right]$
21:      **end for**
22: **end for**

---

# 3 RELATED WORK

**Reinforcement learning algorithms.** Recent years have seen great advances in the field of reinforcement learning due to the use of deep neural networks as function approximators. Mnih et al. (2013b) introduced DQN, an off-policy algorithm learning a parametrized Q-value function through updates based on the Bellman equation. The DQN algorithm only computes the Q-value function, it does not learn an explicit policy. In contrast, policy-based methods such as REINFORCE (Williams, 1992) learn a parameterized policy, typically by following the policy gradient (Sutton et al., 1999) estimated through Monte Carlo approximation of future returns. Such methods suffer from high variance, causing low sample efficiency. Actor-critic algorithms, such as A2C and A3C (Sutton et al., 2000; Mnih et al., 2016), decrease the variance of the estimate by jointly learning policy and value functions, and using the latter as an action-independent baseline for calculation of the policy gradient. The PPO algorithm (Schulman et al., 2017) optimizes a clipped surrogate objective in order to allow multiple updates using the same sampled data.

**Continuous control.** Lillicrap et al. (2015) adapted Q-learning to continuous action spaces. In addition to a Q-value function, they learn a deterministic policy function optimized by backpropagating the gradient through the Q-value function. Haarnoja et al. (2018) introduce Soft Actor-Critic (SAC): a method learning in a similar way, but with a stochastic policy optimizing the Maximum Entropy RL (Levine, 2018) objective. Similarly to our method, SAC also samples from the policy during training.

**Advantage-weighted regression.** The QWR algorithm is a successor of AWR proposed by Peng et al. (2019), which in turn is based on Reward-Weighted Regression (RWR, Peters & Schaal (2007)) and AC-REPS proposed by Wirth et al. (2016). Mathematical and algorithmical foundations of advantage-weighted regression were developed by Neumann & Peters (2009). The algorithms share the same good theoretical properties: RWR, AC-REPS, AWR, and QWR losses can be mathematically reformulated in terms of KL-divergence with respect to the optimal policy (see formulas (7)-(10) in Peng et al. (2019)). QWR is different from AWR in the following key aspects: instead of empirical

returns in the advantage estimation we train a $Q$ function (see formulas 1 and 3 below for precise definition) and use sampling for the actor. QWR is different from AC-REPS as it uses deep learning for function approximation and Q-learning for fitting the critic, see Section 2.

Several recent works have developed algorithms similar to QWR. We provide a brief overview and ways of obtaining them from the QWR pseudocode (Algorithm 2). AWR can be recovered by learning a value function $V(s)$ as a critic (Line 14) and sampling actions from the replay buffer (lines 12 and 18 in Algorithm 2). AWAC (Nair et al., 2020) modifies AWR by learning a Q-function for the critic. We get it from QWR by sampling actions from the replay buffer (lines 12 and 18). Note that compared to AWAC, by sampling multiple actions for each state, QWR is able to take advantage of Q-learning to improve the critic. CRR (Wang et al., 2020) augments AWAC with training a distributional Q-function in Line 14 and substituting different functions for computing advantage weights in Line 20 [1]. Again, compared to CRR, QWR samples multiple actions for each state, and so can take advantage of Q-learning. In a way similar to QWR, MPO (Abdolmaleki et al., 2018) samples actions during actor training to improve generalization. Compared to QWR, it introduces a dual function for dynamically tuning $\beta$ in Line 20, adds a prior regularization for policy training and trains the critic using Retrace (Munos et al., 2016) targets in line 13. QWR can be thought of as a significant simplification of MPO, with addition of Q-learning to provide a better baseline for the actor. Additionally, the classical DQN (Mnih et al., 2013a) algorithm for discrete action spaces can be recovered from QWR by removing the actor training loop (lines 16-21), computing a maximum over all actions in Q-network training (line 13) and using an epsilon-greedy policy w.r.t. the Q-network for data collection.

**Offline reinforcement learning.** Offline RL is the main topic of the survey Levine et al. (2020). The authors state that "offline reinforcement learning methods equipped with powerful function approximation may enable data to be turned into generalizable and powerful decision making engines". We see this as one of the major challenges of modern RL and this work contributes to this challenge. Many current algorithms perform to some degree in offline RL, e.g., variants of DDPG and DQN developed by Fujimoto et al. (2018); Agarwal et al. (2019), as well as the MPO algorithm by Abdolmaleki et al. (2018) are promising alternatives to AWR and QWR analyzed in this work.

ABM (Siegel et al., 2020) is a method of extending RL algorithms based on policy networks to offline settings. It first learns a prior policy network on the offline dataset using a loss similar to Equation 1, and then learns the final policy network using any algorithm, adding an auxiliary term penalizing KL-divergence from the prior policy. CQL (Kumar et al., 2020) is a method of extending RL algorithms based on Q-networks to offline settings by introducing an auxiliary loss. To compute the loss, CQL samples actions on-line during training of the Q-network, similar to Line 14 in QWR. EMaQ (Ghasemipour et al., 2020) learns an ensemble of Q-functions using an Expected-Max backup operator and uses it during evaluation to pick the best action. The Q-network training part is similar to QWR with $F = \max$ in line 13 in Algorithm 2.

The imitation learning algorithm MARWIL by Wang et al. (2018) confirms that the advantage-weighted regression performs well in the context of complex games.

## 4 EXPERIMENTS

**Neural architectures.** In all MuJoCo experiments, for both value and policy networks, we use multi-layer perceptrons with two layers 256 neurons each, and ReLU activations. In all Atari experiments, for both value and policy networks, we use the same convolutional architectures as in Mnih et al. (2013a). To feed actions to the network, we embed them using one linear layer, connected to the rest of the network using the formula $o \cdot \tanh(a)$ where $o$ is the processed observation and $a$ is the embedded action. This is followed by the value or policy head. For the policy, we parameterize either the log-probabilities of actions in case of discrete action spaces, or the mean of a Gaussian distribution in case of continuous action spaces, while keeping the standard deviation constant, as $0.4$.

---

[1]CRR sets the advantage weight function $f$ to be a hyperparameter in $\log \pi_\theta(a_j|s) f(Q_\phi(s, a_j) - \hat{V})$ (Line 20). In QWR, $f(x) = \exp(x/\beta)$.

| Algorithm | Half-Cheetah | Walker | Hopper | Humanoid |
|-----------|-------------|--------|--------|----------|
| QWR-LSE | $2323 \pm 332$ | $\mathbf{1301 \pm 445}$ | $\mathbf{1758 \pm 735}$ | $511 \pm 57$ |
| QWR-MAX | $2250 \pm 254$ | $1019 \pm 1185$ | $1187 \pm 345$ | $503 \pm 49$ |
| QWR-AVG | $1691 \pm 682$ | $1052 \pm 231$ | $420 \pm 65$ | $455 \pm 41$ |
| AWR | $-0.4 \pm 0$ | $67 \pm 11$ | $110 \pm 81$ | $500 \pm 4$ |
| SAC | $\mathbf{5492 \pm 8}$ | $493 \pm 6$ | $1197 \pm 175$ | $\mathbf{645 \pm 27}$ |
| PPO | $51 \pm 41$ | $-14 \pm 98$ | $15 \pm 75$ | $72 \pm 18$ |

Table 1: Comparison of variants of QWR with AWR (Peng et al., 2019), SAC (Haarnoja et al., 2018) and PPO (Schulman et al., 2017) on 4 MuJoCo environments at 100K samples.

| Algorithm | Boxing | Breakout | Freeway | Gopher | Pong | Seaquest |
|-----------|--------|----------|---------|--------|------|----------|
| QWR-LSE | $\mathbf{4.6}$ | $\mathbf{8}$ | $21.2$ | $\mathbf{776}$ | $-7.6$ | $308$ |
| QWR-MAX | $-1.8$ | $0.8$ | $16.8$ | $580$ | $\mathbf{-2}$ | $252$ |
| QWR-AVG | $-0.8$ | $1.4$ | $19.2$ | $548$ | $-9$ | $296$ |
| PPO | $-3.9$ | $5.9$ | $8$ | $246$ | $-20.5$ | $\mathbf{370}$ |
| OTRainbow | $2.5$ | $1.9$ | $\mathbf{27.9}$ | $349.5$ | $-19.3$ | $354.1$ |
| MPR | $16.1$ | $14.2$ | $23.1$ | $341.5$ | $-10.5$ | $361.8$ |
| MPR-aug | $30.5$ | $15.6$ | $24.6$ | $593.4$ | $-3.8$ | $603.8$ |
| SimPLe | $9.1$ | $16.4$ | $20.3$ | $845.6$ | $12.8$ | $683.3$ |
| Random | $0.1$ | $1.7$ | $0$ | $257.6$ | $-20.7$ | $68.4$ |

Table 2: Comparison of variants of QWR with the sample-efficient variant of Rainbow (Hessel et al., 2017; van Hasselt et al., 2019), MPR (Schwarzer et al., 2020), SimPLe (Kaiser et al., 2019) and random scores on 6 Atari games at 100K samples. We report results of the the augmented and on-augmented version of the MPR algorithm. Since MPR and SimPLe are based on learning a model of the environment, we do not consider them when choosing the best scores.

## 4.1 SAMPLE EFFICIENCY

Since we are concerned with sample efficiency, we focus our first experiments on the case when the number of interactions with the environment is limited. To use a single number that allows comparisons with previous work both on MuJoCo and Atari, we decided to restrict the number of interactions to 100K. This number is high enough, that the state-of-the-art algorithms such as SAC reach good performance.

We run experiments on 4 MuJoCo environments and 6 Atari games, evaluating three versions of QWR with the 3 backup operators introduced in Section 2.3: QWR-LSE (using log-sum-exp), QWR-MAX (using maximum) and QWR-AVG (using average). For all those experiments, we set the Q target truncation horizon $T$ to 3 and the number of action samples $k$ to 8. We discuss the choice of these values and show ablations below, while more experimental details are given in Appendix A.1.

In Tables 1 and 2 we present the final numbers at 100K samples for the considered algorithms and environments. To put them within a context, we also provide numbers for SAC, PPO, OTRainbow - a variant of Rainbow tuned for sample efficiency, MPR and SimPLe.

On all considered MuJoCo tasks, QWR exceeds the performance of AWR and PPO. The better sample efficiency is particularly well visible in the case of Walker, where each variant of QWR performs better than any baseline considered. On Hopper, QWR-LSE - the best variant - outpaces all baselines by a large margin. On Humanoid, it comes close to SAC - the state of the art on MuJoCo.

QWR surpasses PPO and Rainbow in 4 out of 6 Atari games. In Gopher and Pong QWR wins even against the augmented and non-augmented versions of the model-based MPR algorithm.

## 4.2 ABLATIONS

In Figure 2 we provide an ablation of QWR with respect to the backup method $F$, multistep target horizon $T$ ("margin") and the number of action samples $k$ to consider when training the actor and the critic. As we can see, the algorithm is fairly robust to the choice of these hyperparameters.

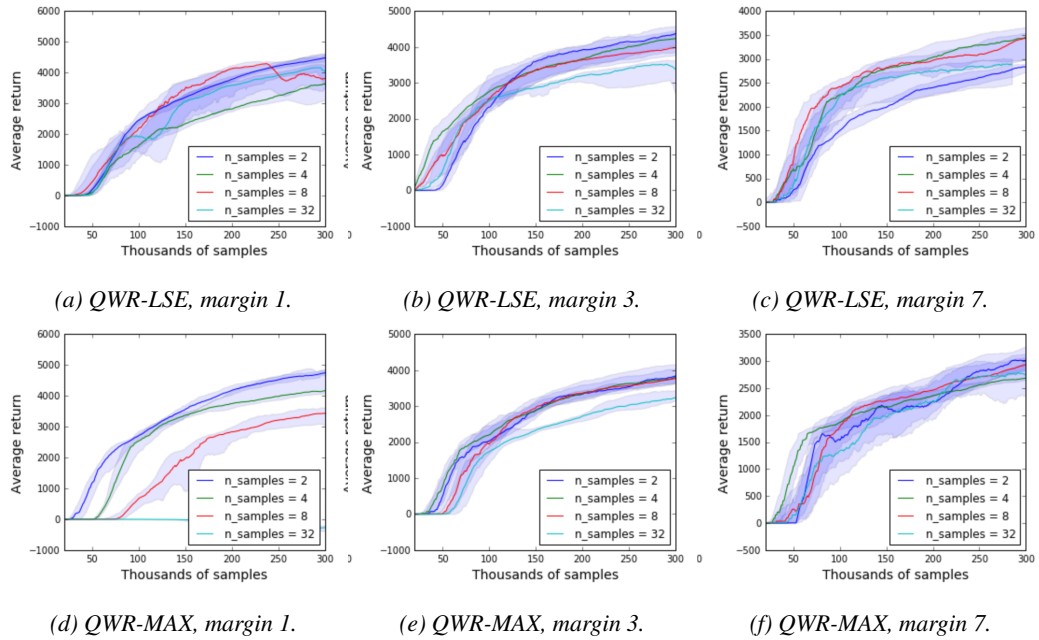

*(a) QWR-LSE, margin 1.*     *(b) QWR-LSE, margin 3.*     *(c) QWR-LSE, margin 7.*

*(d) QWR-MAX, margin 1.*     *(e) QWR-MAX, margin 3.*     *(f) QWR-MAX, margin 7.*

*Figure 2: Ablation of QWR with respect to the margin, the number of action samples and the method of training the critic. The results are shown on the Half-Cheetah environment. The plots show the median of 5 runs with the shaded area denoting the interquartile range.*

In total, the log-sum-exp backup (LSE) achieves the best results – compare Figure 2b and Figure 2e. Max backup performs well with margin 1, but is more sensitive to higher numbers of samples – compare Figure 2d and Figure 2e. The log-sum-exp backup is less vulnerable to this effect – compare Figure 2a and Figure 2d. Higher margins decrease performance – see Figure 2c and Figure 2b. We conjecture this to be due to stale action sequences in the replay buffer biasing the multi-step targets. Again, the log-sum-exp backup is less prone to this issue – compare Figure 2c to Figure 2f.

### 4.3 OFFLINE RL

Both QWR and AWR are capable of handling expert data. AWR was shown to behave in a stable way when provided only with a number of expert trajectories (see Figure 7 in Peng et al. (2019)) without additional data collection. In this respect, the performance of AWR is much more robust than the performance of PPO and SAC. In Figure 3 we show the same result for QWR – in terms of re-using the expert trajectories, it matches or exceeds AWR. The QWR trainings based on offline data were remarkably stable and worked well across all environments we have tried.

For the offline RL experiments, we have trained each algorithm for 30 iterations, without additional data collection. The training trajectories contained only states, actions and rewards, without any algorithm-specific data. In QWR, we have set the per-step sampling policies $\mu$ to be Gaussians with mean at the performed action and standard deviation set to $0.4$, as usual in MuJoCo experiments.

### 5 DISCUSSION AND FUTURE WORK

We present Q-value Weighted Regression (QWR), an off-policy actor-critic algorithm that extends Advantage Weighted Regression with action sampling and Q-learning. It is significantly more sample-efficient than AWR and works well with discrete actions and in visual domains, e.g., on Atari games. QWR consists of two interleaved steps of supervised training: the critic learning the Q function using a predefined backup operator, and the actor learning the policy with weighted regression based on multiple sampled actions. Thanks to this clear structure, QWR is simple to implement and debug. It is also stable in a wide range of hyperparameter choices and works well in the offline setting.

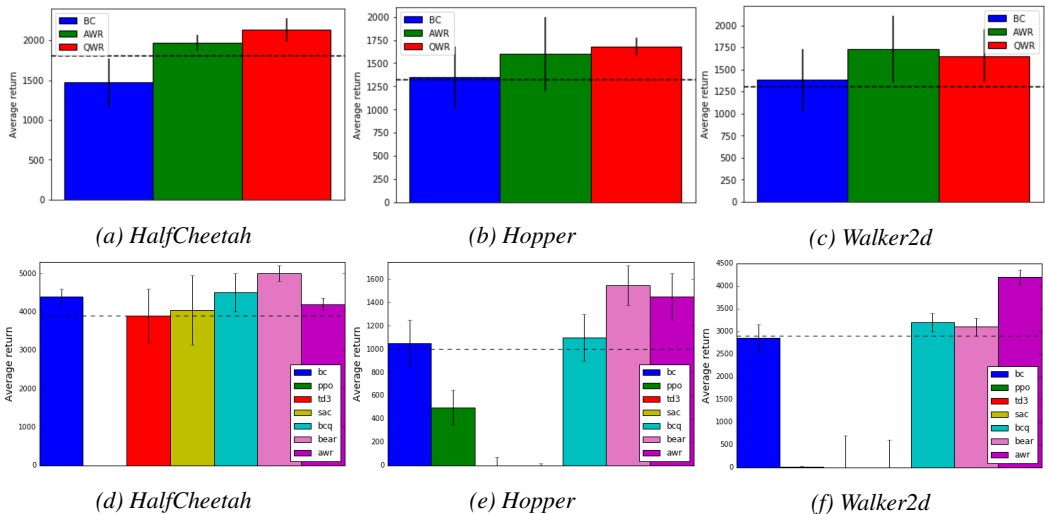

*Figure 3: Figures 3a, 3b and 3c show offline trainings based on 50 trajectories of length 1000 collected by diverse policies. The horizontal lines mark the average return of a policy from the dataset. The bars denote median returns out of 4 runs, and the vertical lines denote the interquartile range. Data for figures 3d, 3e and 3f is borrowed from Peng et al. (2019) to cover a broader family of algorithms and show that offline training fails for many RL algorithms.*

Importantly, we designed QWR thanks to a theoretical analysis that revealed why AWR may not work when there are limits on data collection in the environment. Our analysis for the limited data regime is based on the *state-determines-action* assumption that allows to fully solve AWR analytically while still being realistic and indicative of the performance of this algorithm with few samples. We believe that using the *state-determines-action* assumption can yield important insights into other RL algorithms as well.

QWR already achieves state-of-the-art results in settings with limited data and we believe that it can be further improved in the future. The critic training could benefit from the advances in Q-learning methods such as double Q-networks (van Hasselt et al., 2015) or Polyak averaging (Polyak, 1990), already used in SAC. Distributional Q-learning Bellemare et al. (2017) and the use of ensembles like REM Agarwal et al. (2020) could yield further improvements.

Notably, the QWR results we present are achieved with the same set of hyperparameters (except for the network architecture) both for MuJoCo environments and for Atari games. This is rare among deep reinforcement learning algorithms, especially among ones that strive for sample-efficiency. Combined with its stability and good performance in offline settings, this makes QWR a compelling choice for reinforcement learning in domains with limited data.

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
