# OpenReview forum: "Q-Value Weighted Regression: Reinforcement Learning with Limited Data"
_ICLR.cc/2021/Conference — Reject_

### Official Review · AnonReviewer2 · 2020-10-21
**Review of "Q-value weighted regression: reinforcement learning with limited data"**

**Rating:** 5
**Confidence:** 4

**Review:**

Summary: This paper presents a Q-value weighted regression (QWR) on top of the advantage weighted regression (AWR) to improve the sample efficiency for offline RL settings. Through the analysis to the AWR, the authors claim that it performs poorly in scenarios with discrete actions, which motivates the development of QWR. Empirically, the authors show that QWR is comparable to SAC in continuous tasks and a variant of Rainbow in discrete tasks.

I think the topic investigated in this work is critical. Particularly in many real-world applications, which are offline, the sample efficiency stands to be the most important problem. The paper is also easy to follow and includes thorough investigation on the literature survey. However, the authors need to pay attention to a few major issues that discourage me to give a decent score.

1. Though this work proposes a new offline RL method, the novelty is marginal. As QWR is completely on top of AWR, which seems quite straightforward. This just looks like an incremental step. The authors mention in the paper they theoretically analyze the AWR to motivate the QWR, but after checking the Appendix, no formal theoretical results have been reported, particularly for the proposed QWR. It would be better to see formally how QWR improves on top of AWR, in terms of returns and some constants.

2. The state-determines-action assumption is problematic. In the paper, the authors say “But in the case of limited data, when only a few trajectories were collected, this assumption may hold, at least for a large subset of the replay buffer, which makes it relevant to the study of sample efficiency.” Such an assumption is quite strong in the paper. Since the authors leverage this assumption to give Theorem 1 for AWR, which reveals shortcomings for AWR. This sets a nice motivation for QWR. However, without this assumption, the motivation becomes weak? How to justify this assumption? I would suggest the authors to show in-depth analysis for AWR and QWR to theoretically show their fundamental distinctions.

3. The experimental results are not convincing. In Table 1, though QWR outperforms both AWR and PPO, but not SAC. Similarly, in Table 2, still QWR cannot outperform completely baselines. Though in the abstract, the authors have mentioned, in the result section, they failed to give any further discussion. For offline RL, AWR completely outperforms the proposed QWR, though they are close. Still, the authors just directly described the results without discussion.

*****************************
After reading carefully the rebuttal from the authors, I raise the score a bit as it somewhat clarifies some confusion. However, the paper still needs improvement, particularly in terms of analytical results.

---

> ### Author Response · Authors · 2020-11-17
> **Re: Review of "Q-value weighted regression: reinforcement learning with limited data"**
>
> We thank the reviewer for their detailed and constructive comments which motivated us to significantly improve the paper.
>
> Particularly, we thank the reviewer for pointing out the lack of theoretical results for QWR. We have added a theoretical analysis of the algorithm - Appendix A.3 in the new revision. The conclusion from our analysis is that while AWR can easily fail and repeat actions stored in the replay buffer, QWR learns a smoothed policy, which retains probability mass on the other actions. We verify this theoretical result experimentally on a simple task, as described in the new Section 2.2. The toy task illustrates the key problem with a family of RL algorithms that contains AWR and, together with the new Theorem 2 in Appendix A.3, demonstrates how QWR solves the issue.
>
> As for the state-determines-action assumption, we find it well-justified in both continuous control problems and any real-world task that takes high-dimensional input with even a slight noise. A continuous policy in continuous state space will almost never (with probability 0) return to a previously-visited state, nor will it visit it again in a new trajectory. Similarly, in a high-dimensional real-world task, such as a robot using an RGB camera, it is improbable that the exact RGB value of every single pixel will ever be repeated - even in the same room at the same light conditions one would expect at least one pixel to be different. Any replay buffer sampled in such settings will satisfy the state-determines-action assumption, except in very rare circumstances. We explained this more clearly in the revised version of the paper and added a toy task in Section 2.2 to illustrate it.
>
> As for our experimental results, we updated the offline RL part as the previous runs used the max backup and not the log-sum-exp backup that we use by default (see the answer to AnonReviewer4 for more details). We believe that the online results are also strong: in the default MuJoCo setting, where SAC is a highly-tuned state-of-the-art baseline, QWR still outperforms it on 2 out of 4 environments and gets outperformed on the other 2, so we consider it to be on par with SAC in this setting. But QWR outperforms all other algorithms on all 4 environments and achieves competitive performance in the offline setting, where SAC does not work at all (as demonstrated by [1]). Most notably, QWR achieves these results with the same hyper-parameters that are used for Atari games, a very different setting where neither SAC nor AWR were demonstrated to achieve positive results. To our best knowledge, no other model-free method to date has been shown to yield such good results on both MuJoCo and Atari, as well as in the offline setting.
>
> [1] Peng et al, 2019 - Advantage-Weighted Regression: Simple and Scalable Off-Policy Reinforcement Learning

---

### Official Review · AnonReviewer1 · 2020-10-28
**Official Blind Review #3**

**Rating:** 6
**Confidence:** 3

**Review:**

This paper focuses on offline policy learning with a limited dataset and proposes a sample efficient algorithm called Q-Value Weighted Regression. Based on the Advantage Weighted Regression algorithm, this algorithm calculates the advantage of the sampling policy \mu by estimated Q-value function. Experiment results show that the QWR algorithm has better performance than the AWR algorithm with limited data.
This paper is well-written and easy to follow.  The main contribution is delivered:  New sample efficient algorithm. However, I still have some concerns about this paper.
First, in the experiment part, the authors compared the performance between three kinds of QWR algorithms and the AWR algorithm. Though QWR has better performance than the AWR algorithm, it seems that the QWR algorithms' output is more unstable than other algorithms.
Second, as the authors mention in related work, several recent works have developed algorithms similar to the QWR algorithm. It is better to show the performance of those algorithms in the experiment. If those developed algorithms have similar performance to the QWR algorithm, the QWR algorithm's importance may be affected.
Third, in the critic part of the algorithm (line 7 to line 15), the Q^*-value function is estimated by sample a’ and the Q_{\phi} value function uses samples (independent sample for two value function). However, in the actor part of the algorithm (line 16 to line 20), both the value function Q_{\phi} and \hat{V} is estimated by samples (same sample for two value function). So, I was wondering why there is a difference between the critic-step and actor-step?
Finally, in the formula (5), it seems strange that there is an expectation for random variable a’ and variable s', but none of them appear in the formula.

---

> ### Author Response · Authors · 2020-11-17
> **Re: Official Blind Review #3**
>
> We thank the reviewer for their insightful comments.
>
> About the formulas in the algorithm: in the actor loop, the a_i actions are actions sampled by the policy mu in state s, sampled from the replay buffer. In the critic loop, the a’_i actions are sampled by mu in state s’, which is the state after s. Those actions are used to compute the target using a predefined backup operator, using the current reward r and a bootstrapped value from the next state s’. In line 14, Q_\phi(s, a) is the output of the optimized network, that we backpropagate through, while Q^* is the computed target.
>
> We do not include detailed experimental comparison with parallel works derived from AWR as they are new to us. We have instead pursued a different direction in our experiments: we show competitive results in terms of sample-efficiency on Atari - a challenging benchmark with discrete actions, not included in those works. It is a remarkable feature of QWR - as far as we know not repeated by any RL algorithm so far, including the new works - that it works and is sample-efficient with the same hyper-parameters (without any additional tuning) on environments as diverse as Atari games and MuJoCo, as well as in the offline setting.
>
> We also thank the reviewer for pointing out the confusing notation in Formula (5). The random variable s’ was supposed to be sampled from the environment’s transition operator, and a’ was not supposed to be bound at all - we have corrected this in the revised version of the paper.

---

### Official Review · AnonReviewer4 · 2020-10-28
**Large problems remain unaddressed from previous submission**

**Rating:** 3
**Confidence:** 4

**Review:**

This paper proposes a variant of AWR with an added Q-function. It is motivated by the *state-determines-action* assumption, which is used to argue that AWR should always return the data-collecting policy if optimized to convergence. Experimental evaluation of QWR focuses on online sample efficiency and offline performance.

I have reviewed this paper for a previous conference. It seems that the issues pointed out during that review process have not been addressed. Most importantly, the main motivating theorem (Theorem 1) for QWR appears to be incorrect: it relies on $\log \pi(a \mid s)$ being nonpositive, but this is not true in general in continuous action spaces. Instead of fixing the theorem, it has been moved to the appendix.

The algorithm presented here is largely the same as that in two recent papers: [advantage-weighted actor critic](https://arxiv.org/abs/2006.09359) and [critic regularized regression](https://arxiv.org/abs/2006.15134). It would be unfair to fault this paper for its similarity to others written in the same timeframe, but it does probably mean that the paper should be judged more on its experimental evaluation and analysis than on its novelty. By the metric of experimental evaluation, it falls short. The offline setting has recently seen more standardization of benchmarks (see, eg, [D4RL](https://arxiv.org/abs/2004.07219) and the [DQN Replay Dataset](https://offline-rl.github.io/)), and evaluating on these would allow for cleaner comparison to other works. (To the paper's credit, it does discuss these prior works and their relation to QWR.)

It is possible this comparison was not performed because the offline setting studied is different than the usual offline setting: the data-collecting sampling policy $\pi(a \mid s)$ is required for the optimization in Equation 3, unlike most offline RL works which use only logged trajectories. It is also not necessarily an issue to study a different setting, but it requires some amount of justification for the modified problem setting. It also makes the evaluation more important, because it becomes unclear whether any gains are due to improving the algorithm or relaxing the problem statement.

If I am mistaken about the main theorem being incorrect, I will happily increase my score.

---

> ### Author Response · Authors · 2020-11-17
> **Re: Large problems remain unaddressed from previous submission**
>
> We thank the reviewer for the detailed insights and suggestions.
>
> As for the main objection of the Reviewer: we believe that the statement of Theorem 1 is correct and we have included an extended discussion in the revised version of the paper. The confusion comes from the use of probability density functions in the continuous case vs probability functions in the discrete case. And indeed - logarithms of probability density functions can take arbitrary values, but the maximum will still be reached by the Dirac delta focused on the action from the buffer, so a policy placing probability 1 on action a, as stated in the Theorem. The Dirac delta is not a policy expressible by a neural network (it’s not even a function) though. In the updated revision of the paper we present the proof of the theorem for the case most relevant to RL experiments, when the policy is a Gaussian. We show that indeed the mean converges to the specified action and the variance to 0 (so -log pi(a) -> infty). We hope that the arguments added in the revision in Appendix A.3 are sufficient to convince the reviewer about the theorem - and we will be grateful for further questions if this is not the case.
>
> We also thank the reviewer for pointing out that our offline evaluation setting is different than the usual, by including the parameters of the sampling policy in the logged trajectories. This has been our mistake. We have rerun our offline experiments in the correct setting, without any extra information stored in the trajectories. The results are included in the new version of the paper. Furthermore, the old offline results were reported for an old version of our method, using the max backup - the new results, with the log-sum-exp backup, are reported in the revised version and they are in fact better than the previous ones.
>
> We indeed did not manage to evaluate our method on the new offline RL datasets as we do not have the setup ready for them yet. We will try to do that as soon as we can, but we also believe that the competitive results we achieve in terms of sample-efficiency on online Atari - a challenging benchmark with discrete actions - are a convincing point for QWR performance.

---

> > ### Comment · AnonReviewer4 · 2020-11-17
> > **Re: Large problems remain unaddressed from previous submission**
> >
> > Thanks for your comment. To get to the heart of the matter, the statement in question is:
> > > Recall that the number $\alpha_\mathcal{D}^{\mathbf{s},\mathbf{a}}$ is non-negative as it is an exponent of another number. Thus the value $\log \pi(\mathbf{a} \mid \mathbf{s}) \alpha_\mathcal{D}^{\mathbf{s}, \mathbf{a}}$ can be at most $0$, since the logarithm of a probability is negative or $0$.
> >
> > Take the policy to be $\text{Uniform}(0, \frac{1}{2})$ for all states, and let $\mathbf{a}$ be anything within its boundaries. What is $\log \pi(\mathbf{a} \mid \mathbf{s})$?

---

> > > ### Author Response · Authors · 2020-11-18
> > > **Re: Large problems remain unaddressed from previous submission**
> > >
> > > We are grateful for the comment, we did not make it clear enough. The first part of the proof of the Theorem that contains the statement in question (log pi is non-negative) indeed applies only to discrete action spaces (clarified in the current revision). The Theorem is true for arbitrary action spaces though - so also for continuous distributions - it needs a different proof, a sketch of which we added in the revision in Appendix A 2.1. In particular, we prove it there for the family of  Gaussian distributions, which is commonly used in RL research. We clarify there that if pi were a continuous probability distribution (as in the discrete case) then pi(a|s) would be 0 and its logarithm undefined. But in the continuous case we actually operate on pi being the probability density function and then log pi(a|s) can indeed be positive. The theorem still holds though. Please take a look at the proof for the Gaussian case in A 2.1 - it does not use the statement in question.
> > >
> > > Here is a short version for reference. In the case of Gaussian policies, the PDF $\pi(\mathbf{a|s})$ is characterized by $\sigma(s)$ and $\mu(s)$ of the Gaussian in state $s$, that is
> > > $$ \pi(\mathbf{a|s}) = \frac{1}{\sigma\sqrt{2\pi}} e^{-\frac{1}{2}(\frac{a-\mu}{\sigma})^2}$$
> > > So $\log \pi(a|s) = -\log(\sigma\sqrt{2\pi}) - \frac{1}{2}(\frac{a-\mu}{\sigma})^2$.  This is maximized by choosing $\mu = a$ and reducing $\sigma$ towards $0$, which converges to $\pi$ being the probability distribution that assigns probability 1 to $a$ - a Dirac delta.

---

> > > > ### Comment · AnonReviewer4 · 2020-11-19
> > > > **Re: Large problems remain unaddressed from previous submission**
> > > >
> > > > If the theorem is stated in terms of general policy parameterizations with no restrictions, it would be better to prove the general result.
> > > >
> > > > I am not actually sure the statement is true for all policy families; it seems to rely on a Dirac delta (at all locations) being a limiting case of the distribution in question. Can you make this argument for, say, exponential distributions with the advantages $\alpha$ being largest at some positive constant? It is possible it is still true, but given that this is the motivating theorem of the paper, it seems important to show why the potential counterexamples are not in fact counterexamples.
> > > >
> > > > The added discussion in A.2.1 shows that the mode of a Gaussian is at its mean. That certainly sounds correct to me! But this is also a standard exercise in introductory probability, and I’m not sure it completely bolsters the argument, at least given how the theorem is stated and used in the main paper.
> > > >
> > > > I think the review still holds. For future submissions, the paper would benefit from the following changes (in addition to fixing the aforementioned issues with theorem 1):
> > > > 1. How does the state-determines-action assumption interact with function approximation? It is true that in continuous state spaces you are unlikely to ever see the exact same state twice, but you will see plenty of very similar states. If you have two nearly identical states in the buffer with different corresponding actions, ie $(\mathbf{s}, \mathbf{a}_1), (\mathbf{s}+\epsilon, \mathbf{a}_2)$, and are using a restricted (eg, linear) policy class, what happens? This setting is probably closer to the truth in terms of explaining the behavior of AWR.
> > > > 2. On the empirical end, if you want to compare offline performance, doing so with standardized offline datasets (linked in the review) will make the results more interpretable and reproducible by others.

---

> > > > > ### Author Response · Authors · 2020-11-20
> > > > > **Re: Large problems remain unaddressed from previous submission**
> > > > >
> > > > > We are grateful that the Reviewer recognizes that Theorem 1 is true in the case of discrete distributions and the case of Gaussian distributions. While we believe that it is true in general for continuous distributions where mean and variance can be controlled, let us point out that the two cases above cover all experiments from our paper and all experiments from all our references too -- in fact a significant majority of deep RL experiments in the literature falls into those categories.
> > > > >
> > > > > Just to address the Reviewer’s concern, here is how the theorem would work for the family of two-sided shifted exponential distributions (the Laplace distributions, to cover not only positive numbers) with PDF $g(\lambda, m) = (\lambda/2) e^{-\lambda |a - m|}$. We can first differentiate $g$ with respect to $m$, $\frac{\partial{}}{\partial {m}}g(\lambda_0, m_0)  = \pm \frac{\lambda |m-a|}{2} e^{-\lambda |a - m|}$. Hence $\frac{\partial{}}{\partial {m}}g(\lambda_0, m_0) = 0$ if and only if $m=a$. The distribution $g(\lambda, a)$ reduces to $\lambda/2$ and the maximum is obtained for $\lambda\to\infty$. In the limit we are getting the Dirac distribution, as stated in the Theorem.
> > > > >
> > > > > As for the new suggestions of the Reviewer:
> > > > > In our paper we study deep reinforcement learning, so we consider the study of linear RL beyond the scope of the paper - all our experiments and most references are on deep RL. Neither the original AWR paper nor any of the AWR follow-up papers we know of studies the linear setting, so we challenge the claim that the linear setting is in any way closer to the truth - in fact to our knowledge it has not been successfully used with AWR at all yet. On the other hand, neural networks even with just 2 layers are powerful approximators capable of separating even very similar states. For quick verification, we created a thousand random uniformly distributed 32-dimensional vectors in [-1, 1] and added random uniform noise from [-0.01, 0.01] to corrupt them. A 2-layer MLP with 512 hidden neurons and a ReLU non-linearity reaches over 98% accuracy in distinguishing the original vectors from the corrupted ones each time (we ran it 3 times). Should we include this experiment in the revised version of the paper? We omitted it as we believed that this ability of neural networks to memorize even small differences in the training data is well-known - see e.g. the classical result of [1] and the more recent works [2], [3], studying the memorization and generalization power of DNNs, also in presence of input and label corruption. Though maybe it is worth adding the experiment we mentioned to clarify that the state-determines-action assumption is indeed relevant to standard deep RL settings in real-world experiments.
> > > > > As for the offline experiments, we used a similar setting as the AWR work - training the agents on the same sets of precollected trajectories from MuJoCo tasks. Since we were not aware of the other datasets before, we are not able to run such experiments within the short response period. However, we do plan to include them in the final version of the paper.
> > > > >
> > > > > We want to stress though that while the theorem we discuss motivates our paper, it is not the main point of our work. We consider our positive results -- the introduction of QWR, the fact that QWR yields state-of-the-art experimental results while being stable and sample-efficient, and Theorem 2 with positive results about QWR even in the state-determines-action setting -- as more important than the negative results from the theorem about AWR. We hope that with the above replies we have addressed the issues presented by the Reviewer, and we are of course happy to address any further concerns.
> > > > >
> > > > > [1] Cybenko - Approximation by Superpositions of a Sigmoidal Function (1989)
> > > > >
> > > > > [2] Nakiran et al. - Deep Double Descent: Where Bigger Models and More Data Hurt (2019)
> > > > >
> > > > > [3] Zhang et al. - Understanding deep learning requires rethinking generalization (2017)

---

### Official Review · AnonReviewer3 · 2020-11-02
**Not novel enough**

**Rating:** 4
**Confidence:** 3

**Review:**

Summary: The paper proposes an off-policy actor-critic QWR algorithm that extends the AWR from (Peng et al. (2019)). The QWR algorithm estimates the Q-function using parameterized Q-network whereas AWR estimates the V-function.

I think the paper is not novel enough to guarantee acceptance. The contribution of the paper of the paper is limited given the existing work on AWR.

The Section 2.2 is very confusing. They prove that AWR convergences to a policy that takes the actions appeared in the replay buffer in state-determines-action assumption holds. I think no-algorithm can learn if the state-determines-action assumption holds. Many existing off-policy papers make assumption on sampling policy (strong ignorability (1)).

1. Shalit, Uri, Fredrik D. Johansson, and David Sontag. "Estimating individual treatment effect: generalization bounds and algorithms." International Conference on Machine Learning. PMLR, 2017.

---

> ### Author Response · Authors · 2020-11-17
> **Re: Not novel enough**
>
> We thank the reviewer for their comments that motivated us to clarify many points and expand the analysis.
>
> The difference between our proposed method and AWR is more than using Q-functions instead of V-functions. The reviewer’s summary is closer to AWAC [1], a parallel work. QWR (1) modifies the actor loss by sampling multiple actions for each state in the replay buffer, and (2) trains the Q-network using a procedure similar to Q-learning, in order to provide a better baseline for the actor. The differences between QWR, AWR, AWAC and other related algorithms are discussed in Section 3 of our paper.
>
> We challenge the claim that “no-algorithm can learn if the state-determines-action assumption holds”. In particular, we include in the revised version of the paper a toy environment which has this property with high probability. We show both theoretically and experimentally that AWR fails in this environment, while QWR is capable of solving it - see Section 2.2. In many problems with large enough state spaces (e.g. continuous control tasks or real-world camera inputs) a stochastic agent will almost never return to a previously-visited state, nor will it visit it again in a new trajectory. Any replay buffer sampled in such a task will satisfy the state-determines-action assumption, perhaps except for the initial state.  Still, a number of RL algorithms successfully work in such domains, not only QWR but also SAC [2] and MPO [3]. We believe that the state-determines-action assumption is a way to understand the fundamental differences between RL algorithms and a contribution in its own right. We expanded the analysis of this assumption in Appendix A.2 and added Section A.3 with a proof of a theorem showing the advantages of QWR over AWR.
>
> [1] Nair et al, 2020 - Accelerating Online Reinforcement Learning with Offline Datasets
>
> [2] Haarnoja et al, 2018 - Soft Actor-Critic: Off-Policy Maximum Entropy Deep Reinforcement Learning with a Stochastic Actor
>
> [3] Abdolmaleki et al, 2018 - Maximum a Posteriori Policy Optimisation

---

### Decision · Program_Chairs · 2021-01-07
**Final Decision**

**Decision:**

Reject

**Comment:**

This paper proposed Q-value-weighted regression approach for improving the sample efficiency of DRL. It is related to recent papers on advantage-weighted regression methods for RL. The approach is interesting, intuitive, and bears merits. Developing a simple yet sample-efficient algorithm using weighted regression would be a critical contribution to the field. The work has the potential to make an impact, if it has all the necessary ingredients of a strong paper.

However, reviewers raised a few issues that have to be addressed before the paper can be accepted. As some reviewers pointed out, there seem to be unaddressed major issues from previous submissions. Novelty appears limited, especially because the proposed approach is very similar to recent works (e.g., AWR). The experiment section lacks comparison to recent similar algorithms, and the available comparisons appear to be not strong enough to justify merits of the proposed algorithm. Theorem 1 requires an unrealistic state-determines-action assumption for the replay buffer. Although the authors made an effort to justify this assumption, it remains very problematic and rules out most randomized/exploration algorithms.